Use of open mobile mapping tool to assess human mobility traceability in rural offline populations with contrasting malaria dynamics

Carrasco-Escobar Gabriel gabriel.carrasco@upch.pe 1 2
Castro Marcia C. 3
Barboza Jose Luis 1
Ruiz-Cabrejos Jorge 1
Llanos-Cuentas Alejandro 4
Vinetz Joseph M. 4 5
Gamboa Dionicia 1 4 6
1 Laboratorio ICEMR-Amazonia, Laboratorios de Investigación y Desarrollo, Facultad de Ciencias y Filosofía, Universidad Peruana Cayetano Heredia , Lima , Peru
2 Division of Infectious Diseases, Department of Medicine, University of California, San Diego , La Jolla , CA , United States of America
3 Department of Global Health and Population, Harvard T.H. Chan School of Public Health , Boston , MA , United States of America
4 Instituto de Medicinal Tropical Alexander von Humboldt, Universidad Peruana Cayetano Heredia , Lima , Peru
5 Department of Infectious diseases, School of Medicine, Yale University , New Haven , CT , United States of America
6 Departamento de Ciencias Celulares y Moleculares, Facultad de Ciencias y Filosofía, Universidad Peruana Cayetano Heredia , Lima , Peru
Blackburn Jason
Electronic publication date: 2019 Jan 22
Publication date: 2019
Volume: 7
Electronic Location ID: e6298
Received 2018 Oct 23; Accepted 2018 Dec 18
Copyright: ©2019 Carrasco-Escobar et al.
Copyright year: 2019
Copyright holder: Carrasco-Escobar et al.
License: This is an open access article distributed under the terms of the Creative Commons Attribution License, which permits unrestricted use, distribution, reproduction and adaptation in any medium and for any purpose provided that it is properly attributed. For attribution, the original author(s), title, publication source (PeerJ) and either DOI or URL of the article must be cited.
License URL: https://creativecommons.org/licenses/by/4.0/

Keywords: Amazon, Human mobility, Contact network, Malaria, Network, Infectious diseases, Epidemics

Funding: NIH/Fogarty International Center Global Infectious Diseases Training Program D43 TW007120 NIH-NIAID U19AI089681 Gabriel Carrasco-Escobar was supported by NIH/Fogarty International Center Global Infectious Diseases Training Program (D43 TW007120). This work was funded by NIH-NIAID (U19AI089681) to Joseph M. Vinetz. The funders had no role in study design, data collection and analysis, decision to publish, or preparation of the manuscript.

==============================
Infectious disease dynamics are affected by human mobility more powerfully than previously thought, and thus reliable traceability data are essential. In rural riverine settings, lack of infrastructure and dense tree coverage deter the implementation of cutting-edge technology to collect human mobility data. To overcome this challenge, this study proposed the use of a novel open mobile mapping tool, GeoODK. This study consists of a purposive sampling of 33 participants in six villages with contrasting patterns of malaria transmission that demonstrates a feasible approach to map human mobility. The self-reported traceability data allowed the construction of the first human mobility framework in rural riverine villages in the Peruvian Amazon. The mobility spectrum in these areas resulted in travel profiles ranging from 2 hours to 19 days; and distances between 10 to 167 km. Most Importantly, occupational-related mobility profiles with the highest displacements (in terms of time and distance) were observed in commercial, logging, and hunting activities. These data are consistent with malaria transmission studies in the area that show villages in watersheds with higher human movement are concurrently those with greater malaria risk. The approach we describe represents a potential tool to gather critical information that can facilitate malaria control activities.

Introduction

The process of globalization has expanded the limits of human mobility and connectivity, creating a new dimension for infectious diseases dynamics and spread (Prothero, 1977; Martens & Hall, 2000; Stoddard et al., 2009; Funk, Salathé & Jansen, 2010; Tatem & Smith, 2010). Focused on malaria, human mobility was considered to be a key factor for control and eventually elimination (Pindolia et al., 2012; Wesolowski et al., 2012; Sturrock et al., 2015; Peeters Grietens et al., 2015), encompassing International Tatem & Smith, 2010; Findlater & Bogoch II, 2018, seasonal (Buckee, Tatem & Metcalf, 2017; Wesolowski et al., 2017), and local migration (Searle et al., 2017).

Recently, new technologies have leapfrogged major challenges to collecting reliable mobility data. One successful project was conducted on mobile phone data to assess human mobility (González, Hidalgo & Barabási, 2008; Barabási, 2009) and its application to understand malaria dynamics (Buckee et al., 2013; Wesolowski et al., 2016). Similarly, Google Location History (GLH) was explored as another potential source of human mobility data (Ruktanonchai et al., 2018). Finally, data-collecting wearables such as GPS data-loggers have been proposed to collect fine-scale traceability data (Vazquez-Prokopec et al., 2013; Searle et al., 2017).

However, lack of telephone landline, mobile phone coverage or internet infrastructure prevents the use of approaches using secondary data (i.e., cellular records and GLH) in rural settings, as in the Amazon region. Moreover, dense tree coverage reduces the performance and reliability of GPS data-logger devices (Rempel, Rodgers & Abraham, 1995; Sigrist, Coppin & Hermy, 1999; Danskin et al., 2009). Thus, our study sought to estimate human mobility based on self-reported traceability data collected with a novel open mobile mapping tool, GeoODK. This app provides a suite for offline mapping and visualization of collected geo-referenced data on mobile devices. The main geographical formats in GeoODK are geopoint (point), geoshape (polygon), and geotrace (polyline) that can be associated with other types of information.

Although potential benefits of the use of GeoODK for malaria control was previously demonstrated for household mapping (Fornace et al., 2018), our study explored its geometry mapping features for georeferencing human mobility trajectories. At this micro-geographic scale, mobility data would be most informative to address the exposure to infection (Pindolia et al., 2012) relative to heterogeneous environmental and transmission landscapes (Perchoux et al., 2013), in order to better understand the underlying dynamics in villages with contrasting malaria transmission in rural riverine Peruvian Amazon.

Material and Methods

Ethics

The study was approved by the Ethics Review Board of the Regional Health Directorate of Loreto, Universidad Peruana Cayetano Heredia in Lima and the Human Subjects Protection Program of the University of California, San Diego, USA. IR approval number #101518. Participants were enrolled upon written consent.

Case study

We evaluated the use of GeoODK, an open-source mobile mapping tool, to assess human mobility patterns in rural villages in Loreto Region in the Peruvian Amazon with no connection to either internet or telephone. This study was carried out concurrent to the first survey of the second phase of the NIH-funded International Center of Excellence for Malaria Research (ICEMR) Amazonia project in July 2018. The study area encompasses six villages in two watersheds, the Mazan River (villages of Gamitanacocha, Libertad and Primero de Enero), and Napo River (villages of Salvador, Lago Yurac Yacu and Urco Miraño) (Fig. 1A). This rural setting encompasses primary and secondary forest located north of Iquitos City (capital of Loreto) reachable only by boat transportation (∼2–7 h from Iquitos City). Major landmarks in this area are the river patterns and meanders, strongly influenced by rainfall seasonality. Previous studies have described contrasting malaria epidemiology in both watersheds with complex dynamics related to occupational-related mobility (Parker et al., 2013; Carrasco-Escobar et al., 2017).

Figure 1 Study area in Mazan district, Loreto Region, Peruvian Amazon.

(A) Locale of Gamitanacocha (GC), Libertad (LI), Primero de Enero (PE), Salvador (SL), Lago Yurac Yacu (YY) and Urco Miraño (UM). (B) Polylines (or trajectories) collected with GeoODK, each color represents a participant. (C) Heatmap of transit based on trajectories. Maps were produced using QGIS, and the base maps was obtained from OpenStreetMap (http://www.openstreetmap.org) and OpenTopoMap (http://www.opentopomap.org), under CC BY-SA 3.0.

The first survey of the ICEMR project collected census data, travel records and blood samples (biological data collected were not used for this study). A purposive sampling, proportional to the population size in each community, was carried out based on whether the participant self-reported a trip in the previous month (based on the ICEMR data) and aged 18 years or above. Selected participants were asked to geo-locate the route (a.k.a trajectory) of their recent trip (within a month), either in transit or upon return. All geo-located trajectories were collected in July 2018.

ODK and GeoODK setup

Data collection during the first survey of the ICEMR project was implemented in Open Data Kit (ODK—http://www.opendatakit.org) (Hartung et al., 2010). Briefly, ODK is a flexible open-source suite to collect, store, and manage data in resource-constrained environments. For its part, Geographical Open Data Kit (GeoODK –http://www.geoodk.com) (University of Maryland and International Institute for Applied Systems Analysis, College Park, USA) expands ODK capabilities with a comprehensive set of GIS-related tools. Both are capable of collecting data online and offline. However, GeoODK requires an MBTile-format base map to collect georeferenced features during offline survey.

Both applications were set up using the same Google App Engine server, but applications ran separately on the mobile devices. In this study, Samsung Galaxy Tab A, with 8GB internal memory, Android 5.1, and 7-inch screen tablets were used to better display the base map to participants. A subset of the ICEMR data regarding household geo-reference, socio-demographics and travel records was used in this study. GeoODK and ODK data were linked using common participants’ identifiers (PID) in both applications. In addition, a polyline widget was included in GeoODK to show a blank map where participants drew (in fact, georeferenced) the trail of their last movement outside the village. After a brief introduction to the software and a demonstration of the functions (i.e., zoom, and current location), the participants received the tablet and started the polyline creation process. In case the participants required, the interviewer assisted the polyline creation process based on the participant’s directions.

A simpler workflow to generate the base map was used in comparison to previous studies (Fornace et al., 2018). A georeferenced tiff image was constructed based on public geographical data from OpenStreetMap (http://www.openstreetmap.org) and converted to MBtile using maptiler Desktop v. 9.1–1 (Klokan Technologies GmbH—http://www.maptiler.com). Main landmarks (i.e., river patterns) were validated using available Landsat 8 imagery in the period June–July 2018 (time frame were the travels were taking place). Main villages and river names were added to the base map using QGIS 2.18 (QGIS Geographic Information System, Open Source Geospatial Foundation Project: http://www.qgis.org) for a better orientation of the participants. GeoODK stored a performance log to analyze the start and end time of the complete questionnaire, but also per question.

Spatial processing and statistical analyses

The KML output of the polyline widget in GeoODK was imported to QGIS. Each geometry corresponds to a participant, and PID were stored by default as an attribute. After initial validation of PID and whether all geometries were inside the study area, a shapefile was generated to improve spatial handling. No additional geo-processing was conducted over raw polylines data. Each polyline was transformed into a layer of points with the QChainage plugin 2.0.1 (Macho, 2018) separated by 1m. Finally, a heatmap with a radius of 1m was constructed using the point layers. GeoODK trajectories and travel record data (destinations) in the ICEMR survey were validated with the National register villages (NRV), that contain GPS coordinates for each official village. Fisher’s exact test was used for significance testing of categorical factors between participants included in the ICEMR survey and this study. Spearman’s correlation was used to identify the relationship between the time to complete a GeoODK survey and the trajectory distance recorded per participant. Maps were generated with QGIS. All descriptive analyses and visualizations were produced using R v.3.4.3 (R Core Team, 2017).

Results

The study population comprised 33 adult subjects between 19–68 yrs of age (mean = 41 yrs), who had lived in the study site for 6 months to 68 yrs (mean = 26 yrs). There was no refusal to participate in the study. Most participants were male (67%); 12% were illiterate. The most common occupation was farming 60%; 21% also had fished or hunted during the previous month. Inhabitants reported 1–10 trips per month (mean = 2 trips/month). Most trips were recreational—e.g., visit to family or friends (36%), for commerce (21%), or logging/hunting/fishing (21%). Several visits were reported to Mazan (48%) and Iquitos (18%). The time from origin to destination (hereafter known as transit time) ranged from 1–8 hrs (mean = 3 hrs), but 12% of the population spent more than one day in transit, most commonly for logging or hunting. Overall, the total travel time (transit, stay, return) of 45% of the population lasted more than 1 day (mean = 6 days). Of the population who traveled, 73% slept in a house, whereas 27% in the forest or in the boat. The average time between the returning date and the GeoODK data collection was 6.8 days (range: 0–25 days). Importantly, there were no statistically significant differences in age categories (p-value = 0.347), occupation (p-value = 0.305), and travel reasons (p-value = 0.216) between participants in the ICEMR survey with a travel record and aged 18 years or above (n = 233) and this study (n = 33).

Using GeoODK, all geometries (polylines) were correctly mapped to plausible displacement paths (Fig. 1B) along the watercourse, however, in-forest displacements were not possible to validate. All origin and 27 (81.8%) of destination locations were correctly georeferenced, and validated using ICEMR travel records data and NRV data. Six (18.2%) destinations were not available in NRV, most of them logging or hunting areas. Additional information was obtained with GeoODK, such as different routes to Iquitos city (capital of Loreto Region), and locale of non-documented logging areas where inhabitants work frequently. Overall, recorded trajectories ranged from 10–167 km (mean = 42 km), with most of the population (67%) displaced less than 50 km. The transit heatmap based on self-reported trajectories depicts more movements along the Mazan River than the Napo River (Fig. 1C). The total travel time along the Mazan River ranged from 2 hrs to 14 days (mean = 4 days), whereas a lower duration was observed along the Napo River (1 hr to 19 days (mean = 2 days). The average distances of the trajectories of Mazan and Napo River inhabitants were comparable, 48 km and 35 km respectively. High variability was observed in Napo River due to some long trajectories to Iquitos, and also to occupational-related activities in distant areas of the Mazan River. Overall, the average time to complete the geo-referenced data collection in GeoODK was 8 min (range: 1–25 min) per participant, and showed moderate correlation with the distance of the recorded trajectory (Spearman’s rho = 0.565; p-value = 0.002).

Marked mobility patterns were observed in this study (Fig. 2A). The most common patterns were short displacements (7.4–22.1 km) in a short period of time (0–0.3 days) (27%) and mobility within intermediate distances (22.2–66.6 km) and intervals (0.4–2.6 days) (27%). Importantly, the greatest distances and periods were related exclusively to logging and hunting activities (Fig. 2B). Contrasting distributions of travel time and distance were observed among villages (Fig. 2C); however, proximal distributions were observed between villages in the same watersheds (Mazan and Napo). Albeit less marked, a distance/time ratio decay was observed for inhabitants with higher income (Fig. 2D).

Figure 2 Mobility patterns of inhabitants of Mazan district.

(A) Distribution of profiles among categories of trip distance and time (X- and Y-axes in logarithmic scale). (B) Travel patterns per trip reason. (C) Distribution of travel time and distance between villages (villages abbreviations on Y-axis as in Fig. 1). (D) Stratified distribution of the ratio distance/time according to income.

Discussion

In settings with scattered foci of infection, connectivity is a cornerstone for the maintenance of malaria transmission. Although the use of big data and actively data-collecting wearables pave the way for a comprehensive understanding of the role of human mobility in infectious diseases dynamics, these technologies are not available in many of the rural populations that comprise most malaria transmission settings. This study demonstrated the feasibility and the added value of a novel open mobile mapping tool to assess human mobility in rural offline populations. Importantly, this approach allowed for mobility profiling in communities with contrasting malaria transmission and determined, to our knowledge, the first characterization of human mobility patterns in the rural riverine Peruvian Amazon.

The study case setting, Mazan, is classified as a high malaria transmission intensity district by the Ministry of Health (MoH) with an intricate river network where most villages are located. The findings of this study set the stage for the accurate detection and analysis of human mobility patterns in rural riverine settings. The use of GeoODK permitted the collection of detailed trajectories that would have been lost using traditional collection methods. Highlighted benefits include documentation of different routes to the same destination, and precise geo-localization of occupational-related areas. Regarding the former, two main routes to Iquitos city were observed. Trajectories passed through Mazan or Indiana, both commerce-dedicated cities with well-equipped health centers, yet with different malaria incidence rates. Regarding geo-localization of occupational-related areas, there are illegal logging and hunting areas, and no official human settlements that lack georeferenced data (NRV data). Thus this approach complemented the self-reported surveyed data to better understand travel distances and disease exposure of these riverine populations (Parker et al., 2013), that otherwise would not be possible since there is not official GPS registers of these locations.

A recent study demonstrated the accuracy of GeoODK for household mapping, reflecting the spatial orientation of inhabitants when exposed to a blank map (Fornace et al., 2018). Consistently, all reported origins and destinations in the ICEMR survey with available geo-referenced data (NRV data) were correctly mapped with GeoODK. Participants were able to distinguish and georeference their trajectory through different areas within cities or villages that commonly would be lost using structured forms. This is a relevant feature given the fact that there are no addresses in most rural and riverine communities that could be georeferenced, and highly heterogeneous malaria transmission within riverine communities has been reported previously (Carrasco-Escobar et al., 2017).

Although this is a proof-of-concept study, we found interesting aspects of malaria dynamics in this area. From both watersheds, a more intensive travel transit was observed in the Mazan River. Consistently, higher risk of malaria infection (Chuquiyauri et al., 2013; Carrasco-Escobar et al., 2017) and vector exposure (Parker et al., 2013) have been reported in communities along the Mazan River. Population genetic studies in the area detected high heterozygosity and polyclonal infections that were hypothesized to be due to high human mobility (Van den Eede et al., 2010). This intensified malaria dynamic presumably arose from continuous commerce, logging and hunting–related mobility, a factor that was reported as key for greater risk of Plasmodium vivax malaria across Peruvian, Colombian, and the Brazilian Amazon (Sevilla-Casas, 1993; Da Silva-Nunes et al., 2008; Hahn et al., 2014; Carrasco-Escobar et al., 2017) and other malaria settings (Smith et al., 2017). Moreover, the fact that all logging and hunting areas reported by participants were mapped only along the Mazan River, and the distribution of travel time and distances were highly heterogeneous between watersheds (Salonen et al., 2012), supports the idea that malaria control might be addressed at a larger scale, among high-connectivity units.

The data reported here were obtained at a meso-scale (study area extent = 50 km2), but depending on the research question and the base map used in GeoODK, this approach could be conducted at a micro- or macro-scale in a variety of fields. The increased evidence of exophagic biting behavior of Ny. darlingi in the Peruvian Amazon (Reinbold-Wasson et al., 2012; Moreno et al., 2015), urges better understanding of outdoor activities and human mobility patterns to tailor malaria control strategies. In addition, spatial and molecular epidemiology (Delgado-Ratto et al., 2016), among other disciplines, would benefit from accurate human mobility estimates in resource-limited settings. However, longitudinal studies on human mobility are highly recommended. Temporal and seasonal trends of human mobility might better elucidate the underlying malaria exposure (Wesolowski et al., 2012; Tatem et al., 2014; Ruktanonchai et al., 2016; Smith et al., 2017; Wesolowski et al., 2017).

While, encouragingly, the performance of GeoODK to collect human mobility data appear to be reliable regardless of phone network structure or environmental conditions, we find that several validity assessments must be conducted to scale-up its implementation. Firstly, despite a large amount of destinations were possible to validate using travel surveys or National datasets, still remaining the validation of movement data during transit or return. This would be possible outside of forested riverine systems where GPS tracker data, or mobile or GLH data could be used to validate the self-reported trajectories in GeoODK. The use of GPS trackers with survey-grade receivers is suggested as a validation method for travel time and in-forest displacements, since dual-frequency acquisition outperformed other GPS receivers in forested areas. Regrettably, the cost of survey-grade GPS receivers deters its use in population-based epidemiological studies.

Conclusion

In conclusion, a feasible approach to map human mobility traceability in rural villages was presented in this study. Although our findings allowed for the construction of a human mobility framework in the Peruvian Amazon, additional work must be conducted to deepen our understanding of human mobility that could facilitate tailor-made malaria control activities, and mark a turning point for watershed- or mobility-circuit- based control approaches.

We thank Prof. Jan Evelyn Conn and Prof. Kimberly Brouwer for editing and proof-reading the manuscript; Dr. Hugo Rodriguez Ferrucci and Edgar Manrique for their valuable discussions and suggestions.

Additional Information and Declarations

Competing Interests

Author Contributions

Human Ethics

Data Availability

The authors declare there are no competing interests.

Gabriel Carrasco-Escobar conceived and designed the experiments, analyzed the data, prepared figures and/or tables, authored or reviewed drafts of the paper, approved the final draft.

Marcia C. Castro conceived and designed the experiments, authored or reviewed drafts of the paper, approved the final draft.

Jose Luis Barboza performed the experiments, approved the final draft.

Jorge Ruiz-Cabrejos performed the experiments, analyzed the data, approved the final draft.

Alejandro Llanos-Cuentas and Dionicia Gamboa contributed reagents/materials/analysis tools, approved the final draft.

Joseph M. Vinetz contributed reagents/materials/analysis tools, authored or reviewed drafts of the paper, approved the final draft.

The following information was supplied relating to ethical approvals (i.e., approving body and any reference numbers):

The study was approved by the Ethics Review Board of the Regional Health Directorate of Loreto, Universidad Peruana Cayetano Heredia in Lima and the Human Subjects Protection Program of the University of California, San Diego, USA (approval number: #101518).

The following information was supplied regarding data availability:

Shapefiles and metadata are available at figshare:

Carrasco-Escobar, Gabriel (2018): GeoODK_Carrasco-Escobar2018 Shapefile. figshare. Dataset. https://doi.org/10.6084/m9.figshare.7091075.v1

Carrasco-Escobar, Gabriel (2018): GeoODK_Carrasco-Escobar2018 Metadata. figshare. Dataset. https://doi.org/10.6084/m9.figshare.7091078.v1.

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
