# Peer review of "Use of open mobile mapping tool to assess human mobility traceability in rural offline populations with contrasting malaria dynamics"

_PeerJ, doi:10.7717/peerj.6298_

## Round 0.1 · original submission · Minor Revisions

Congratulations on a solid manuscript reporting results of a pilot study. Each of three reviewers has commented that the work and manuscript were well done. While one reviewer suggested a rejection due to lack of novelty, the same reviewer commented on the quality of the work and made recommendations for improvement that should be considered. Namely, previous interviews have been conducted and published in the region. Please review the suggested citation and address this in your response and/or revision.

Of note, you did not report on whether there were refusals to participate. This should be addressed outright in the revision. Beyond these two major points, each reviewer has provided several suggestions to improve the manuscript. I concur with these suggestions and look forward to your revised resubmission soon.

Reviewer 1 ·

Basic reporting

The article is clear and well structured.

Experimental design

The design is solid but lack of originality.

Validity of the findings

Findings are robust but not very new to readers.

Additional comments

The authors used an open mobile mapping tool, GeoODK, to survey the rural riverine Peruvian Amazon, and analyzed travel patterns of participants. My feeling is the methodology design is solid, but it is like the same technique being used again and again in different geographic regions. In other words, it would be a good working report but lacks sufficient novelty for a publication.

1. Given its offline feature, GeoODK seems to be capable of being applied to any population. It has been used in many remote areas in China, Pakistan, and Brazil. It is not surprising it fits the rural riverine population in Amazon. In this sense, this study has no difference to other studies using GeoODK for data collection. Is this worthy of reporting in a publication?

2. The authors claim that their study is the first characterization of human mobility pattern in the rural riverine Peruvian Amazon. Please note that there is some other research that used GIS and semi-structured interviews to understand the same region.

Salonen, Maria, et al. "Critical distances: comparing measures of spatial accessibility in the riverine landscapes of Peruvian Amazonia." Applied geography 32.2 (2012): 501-513.

3. The authors may add some statistical charts to contrast travel patterns in two different watersheds. There are some text about comparison, but graphs would be more intuitive.

Reviewer 2 ·

Basic reporting

This is a well written study with clearly illustrative figures presenting a useful methodology for collecting movement data using tablet-based applications. Although this is pilot study, the authors are clear on these limitations. The study could benefit from some clarifications on the methodology and additional background on the study site.

Experimental design

In general, the research question and methods are clearly defined but there are a few points which would benefit from further description, particularly around the collection of time data and the validation of GPS data.

Lines 44- 49: Presumably at the scale of this study, the purpose of collecting movement data relates more to the where individuals are exposed to infected mosquitoes? If so, this initial introduction would benefit from an additional sentence on exposure relative to environment and local mobility patterns.

Line 58: no mobile phone coverage within this site?

Lines 59-61: The literature around effects of cloud cover on GPS accuracy is somewhat mixed; this statement should be referenced.

Lines 80-83: This would benefit from some further description of the study site, particularly in respect to land cover and available landmarks (buildings, roads, etc. if applicable).

Line 89: What time frame were the participants asked to geo-locate the trips within? I.e. immediately after returning or several weeks following the trip?

Line 102: What census data was used? Did this include any geo-referenced data (such as locations of participant households?)

Line 108: Was the accuracy of this map affected at all by the high cloud cover? Was there any attempt to ground-truth or verify the accuracy of this map and did any of the landmarks change substantially over time (such as changing river patterns relative to season or rainfall)?

Line 117: How was the movement data validated? Was there any attempt to compare reported movements with actual mapped movements?

Line 132-137: How was the data on the time of travel collected? Are these self-reported travel times and, if so, was there any attempt to assess accuracy of these times?

Lines 138: How were plausible displacement paths determined, particularly for forest movements or other activities with no identifiable roads or rivers?

Validity of the findings

The conclusions of the study are clearly stated, with the limitations of the sampling size and pilot study approach described. To improve these conclusions, further data could be included on how the reported GPS data was validated and how validation data was collected.

Lines 182-184: How did the accuracy of self-reported movements for these areas compare to other movements (such as along rivers, roads and to major cities)?

Lines 189-190: clarify what this geo-referenced data included and how it was collected

Additional comments

Two minor comments on the title and abstract:

Title: I would revise this slightly as “use of an open-source offline mobile mapping tool” to clarify the tool was used offline (rather than the populations).

Abstract: Line 27: typo, should be “GeoODK”

·

Basic reporting

Overall the article was well-written and succinct. I think some information could be added in a few specific places:
- Add information on GeoODK in the introduction--it's clear from the methods how it works (drawing polylines to represent travel routes) but it would be good to have some of this info in the intro. In the Methods I would also like to see a line on how researchers would find this tool to use themselves (ie. a link to a download page)
- Add information on the types of mobility that GeoODK-collected data would be most informative for--i'm thinking of the spatial/temporal axes on Fig 1 in this article: https://malariajournal.biomedcentral.com/articles/10.1186/1475-2875-11-205 (I suspect GeoODK data would be most informative on the bottom left end of the figure, for example). This is addressed to a degree in the discussion but it would be nice to see in the introduction as well.
- re: reporting of the study: Did you have any refusals? Refusal rates or non-refusal across all participants should be noted in a line in the Results.

Specific small comments:
- Lines 50 - 61: The first sentence of the paragraph starting line 58 sounds like it applies to all the mobility data referenced in 50-57, when it really only applies to the mobile phone data and GLH data, whereas lines 59-61 apply to GPS data. Please reword for clarity
- Line 86: If this is a traveler-only subset of the census group, I'd like to see some information on the overall population, to compare how the participants differed in terms of age and occupation from the population at-large.

Experimental design

I would like to see a little more detail on the polyline creation process in the Methods--how did participants record their movements, and were these cleaned up before analysis? I'm looking at Fig 1b that has a line that very neatly follows the river, and it doesn't look like a line that a participant could easily have drawn with their finger on a tablet. (perhaps they did, though--just need detail on how the actual data collection occurred)

Did you record how long did it take for people to record their movements? Did this correlate with the amount of mobility data you got with each individual?

Validity of the findings

Is there any data to validate your travel routes with (ie. travel survey data from the same individuals collected during the census)? I'm concerned that the amount of effort required to record each trip might mean that people exclude shorter trips, and I think the paper should address these sorts of recall-related biases in some way. It's not within the scope of this study to do much, but at least it would be good to note validation as an important next step in the discussion--for example, outside of forested riverine systems, researchers could compare the GeoODK-collected data with personally-collected GPS tracker data, to confirm that GeoODK collection captures a large proportion of trips away from home.

Additional comments

Generally, this was an interesting pilot study for the use of GeoODK for tracing travel routes. The main comments I had involve reporting more information--specifically, the possible refusal rates, the amount of time/effort needed for participants to record travel, and contextualizing the GeoODK data better with other types of mobility data, in terms of relevant temporal and spatial scales.

---

## Round 0.2 · accepted · Accept

Thank you for your attention to the comments and recommendations of the reviewers. Both were fully satisfied with the revision, as was I.

Reviewer 2 ·

Basic reporting

No comment

Experimental design

No comment

Validity of the findings

No comment

Additional comments

The authors have addressed all reviewer queries and I would recommend publication.

·

Basic reporting

No comment

Experimental design

No comment

Validity of the findings

No comment

Additional comments

I'm happy with the responses to my comments.